# Robotic-Enhanced Prosthetic Liners for Vibration Therapy: Reducing Phantom Limb Pain in Transfemoral Amputees

**DOI:** 10.3390/s24155026

**Published:** 2024-08-03

**Authors:** Kacey Roehrich, Mary Goldberg, Goeran Fiedler

**Affiliations:** Department of Rehabilitation Science and Technology, School of Health and Rehabilitation Science, University of Pittsburgh, Pittsburgh, PA 15206, USA; mgoldberg@pitt.edu

**Keywords:** phantom pain, phantom limb pain, vibration therapy, artificial limbs, instrumented suspension liners, smart prosthetics

## Abstract

Phantom limb pain, a common challenge for amputees, lacks effective treatment options. Vibration therapy is a promising non-pharmacologic intervention for reducing pain intensity, but its efficacy in alleviating phantom limb pain requires further investigation. This study focused on developing prosthesis liners with integrated vibration motors to administer vibration therapy for phantom limb pain. The prototypes developed for this study addressed previous issues with wiring the electronic components. Two transfemoral amputees participated in a four-week at-home trial, during which they used the vibration liner and rated their initial and final pain intensity on a numeric rating scale each time they had phantom pain. Semi-structured interviews were conducted to gather feedback following the at-home trial. Both participants described relaxing and soothing sensations in their residual limb and phantom limb while using vibration therapy. One participant reported a relaxation of his phantom limb sensations, while both participants noted a decrease in the intensity of their phantom limb pain. Participants said the vibration liners were comfortable but suggested that the vibration could be stronger and that aligning the contacts could be easier. The results of this study highlight the potential effectiveness of using vibration therapy to reduce the intensity of phantom limb pain and suggest a vibration liner may be a feasible mode of administering the therapy. Future research should address optimizing the performance of the vibration liners to maximize their therapeutic benefits.

## 1. Introduction

The prevalence of limb loss in the United States is increasing. An estimated 2.2 million persons in the United States have limb loss, and that number is predicted to reach 3.6 million by 2050 [1,2]. Diabetes mellitus, infections, and vascular conditions are common causes of amputation. Two common phenomena among amputees are phantom limb sensation (PLS) and phantom limb pain (PLP). PLS describes non-painful sensations felt in the absent limb and is experienced by 80–100% of amputees [3]. PLP, on the other hand, describes painful sensations in the absent limb and is experienced by 60–80% of amputees [3]. PLP is often described as shooting, stabbing, or burning pain and can be experienced daily, weekly, or more infrequently. Despite being a widely experienced phenomenon among amputees, the mechanisms of PLP are still not wholly understood, so there is no single, mechanism-based treatment for PLP.

Current interventions cover a spectrum of non-pharmacologic and pharmacologic therapies and surgical procedures, but the effectiveness of a particular therapy depends on the individual. Mirror therapy is a popular non-pharmacologic treatment that uses a reflection of the limb contralateral to the amputated limb to provide visual-proprioceptive feedback to the brain. Mirror therapy has proven to be effective for some individuals; however, a systematic review of 20 studies concluded that the level of evidence is insufficient to support its use to address PLP [4]. Drugs such as gabapentin, morphine, and amitriptyline are commonly administered for PLP and have been shown to reduce pain intensity. A systematic review categorized nine articles with pharmacologic interventions as moderate-quality evidence [5]. These studies had mixed results about whether the administered drugs produced a statistically significant difference in pain on a visual analog scale and functional independence. Not only can drugs be addictive, but the studies also found that participants often experienced other side effects from the drugs [5]. When conservative, non-invasive treatments are not effective, a surgical procedure such as dorsal-root entry zone lesioning could be an option; however, a systematic review found that the studies included in the review had a lack of specificity and small sample sizes; therefore, confident conclusions on the effectiveness of this procedure could not be drawn [5].

Previous studies have found mechanical stimulation through vibration effective at reducing various types of pain [6,7,8]. Mechanical stimulation through vibration, termed vibration therapy, is based on the gate control theory of pain, which suggests that the dorsal horns of the spinal cord act as gates for the pain signals sent to the nervous system. The activity of A-δ fibers and C-fibers keeps the gates open but stimulating Aβ fibers with vibration triggers an inhibitory response, closing the gates and interrupting the transmission of pain signals to the brain [9]. Compared to other therapy options, vibration therapy is non-invasive, non-pharmacologic, and low-risk as it does not have the side effects that often accompany pharmacologic interventions. While the effects of vibration therapy on PLP have been less researched than other types of acute and chronic pain, participants in three studies reported a decrease in PLP intensity after receiving vibration therapy [10,11,12]. While vibration therapy’s effect on PLP needs further research, there is also a need for further research and development on a device that can administer vibration therapy to a residual limb and be worn within the user’s prosthesis socket.

Flexible silicon- or polyurethane-based liners are commonly used for prosthetic suspension and provide an interesting option as the substrate for integrated hardware, such as vibration motors [13,14]. However, early prototypes created by our work group did not pass a bench test protocol that simulated donning and doffing the liner over its suggested lifecycle, and this prototype generation forewent embedded motors. Prior work made strides toward achieving prosthesis-compatible prototypes; however, the thickness of the prototypes and a method of powering the motors remained an unsolved problem.

The first project aim was to further develop a method of securely embedding vibration motors into a liner, so they remain embedded during the regular use of the liner. The second project aim was to develop a method of powering an instrumented prosthesis liner, such as one with integrated vibration motors, via a power source located on the exterior of the socket without affecting the comfort and fit of the socket. The third project aim was to continue to study the effects of vibration therapy on the intensity of phantom limb pain when delivered via a vibration liner, investigating the hypothesis that vibration therapy delivered via a vibration liner would reduce the intensity of PLP.

## 2. Materials and Methods

### 2.1. Study Design

To pursue the stated project aims, we employed an iterative development and validation approach, advancing the device design according to design criteria that were generated and updated following continuous feedback by members of the target population.

### 2.2. Study Participants

Individuals who work with the Prosthetics and Orthotics program at the institution and previously expressed interest in participating in research studies were contacted by the principal investigator. Ethics approval was granted by the responsible oversight body, and enrolled participants provided informed consent before study activities began. Interviews and meetings with the participants were scheduled as needed and conducted via phone, Zoom, or in person at the laboratory. Throughout the project, participants shared their experiences of PLP, tested the prototypes, and provided feedback that influenced the design criteria and prototype iterations. Two participants were screened, consented, and enrolled in the study. Table 1 includes demographic information of the study population.

### 2.3. Design Criteria

In the 2022 practice analysis of American Board for Certification (ABC)-certified orthotists and prosthetists, prosthetists reported using a roll-on liner for 75% of transtibial patients and 80% of transfemoral patients [15]. Prosthetists also reported using suction suspension with a roll-on liner for 30% of lower extremity amputation patients [15]. Given how common roll-on liners and suction suspension are in clinical practice and that both participants use suction suspension, the design criteria described in Table 2 were centered around maintaining the suspension and comfort of the socket and liner. Additional criteria were added following initial interviews with each participant, during which they shared their experiences and preferences.

### 2.4. Design Approach

Based on participant feedback requesting vibration all around the residual limb, the final prototype had two posterior motors located three centimeters from the distal end of the femur and four pairs of motors located eight to ten centimeters above the distal end of the femur and evenly spaced circumferentially around the liner (Figure 1). The motors used in the final prototype were micro vibrating motors with a diameter of 10 mm, a thickness of 3 mm, and a rated DC voltage range of 2.5–3.8 volts. At the rated voltage, the rotational speed of the motors was 11,000 ± 3000 rpm, which equates to a frequency range of 183 ± 50 Hz [17]. The wires were pulled through a channel tunneled through the silicone layer, and two motors were paired with their wires terminating under a pair of conductive hook-and-loop pads. These conductive hook-and-loop pads served as contact points for making the electrical connection with the battery when using the liner. The motors were embedded in a four-millimeter-deep hole cut into the silicone and sealed with Sil-poxy. This method did not create noticeable, raised bumps on the surface of the liner and protected the wires underneath the fabric layer during donning and doffing. The hole cut for the motors did not puncture through to the inside of the liner, so the silicone that interfaced with the skin remained intact.

Since maintaining suction within the socket was a design criterion, the method of making an electrical connection between the socket and liner could not interfere with the air-tight seal. The final socket prototype (Figure 2) had an arrangement of copper tape on the inside of the socket that aligned with the conductive hook-and-loop on the liner upon donning. On the outside of the socket, the copper tape extended over the edge of the socket, and a pair of wires attached to a JST-PH connector from Japan Solderless Terminals were soldered to the tape. A battery holder that mated with the JST-PH connector was affixed to the socket with Velcro. The battery holder held three AA batteries and included a built-in on/off switch.

To use the vibration liner without donning a socket, a conductive wrap was constructed with felt, Velcro, and copper tape (Figure 3). The wrap included a JST-PH connector which connected to the battery holder after it was detached from the socket.

### 2.5. Bench Testing

The average lifespan of a conventional prosthesis liner is six months, so assuming the liner is donned and doffed once per day, it undergoes roughly 400 donning and doffing cycles during its lifespan [16]. The robustness of the prototype, particularly the method of embedding the motors and wires, was assessed by simulating 400 donning and doffing cycles on a plaster mold. Throughout the testing process, the liner was inspected for tears in the silicone, loosening of motors, and any other changes to the prototype. After every 50 cycles, the embedded motors were tested for functionality.

It is necessary to routinely clean the liner since it contacts the skin. Throughout the construction process and before each participant tested the prototype, the liners were cleaned with a pH-neutral prosthetic cleanser and observed for any damage to the liner or added components.

The participants reported cleaning their sockets less frequently than their liners, so a benchmark of 35 cycles was chosen, assuming the socket is cleaned roughly once or twice a week over about six months. When cleaning the socket, three combinations of cleaning materials were tested to simulate the conditions the participants reported and to test a third, more abrasive, condition. All combinations used dish soap and water, but one combination used a cloth alone (SC), a second used a cloth and included wiping down the socket with isopropyl alcohol after washing (SAC), and a third used a scrub brush in addition to wiping down the socket with isopropyl alcohol after washing (SAB).

### 2.6. Clinical Testing

Per the IRB-approved study protocol, the participants provided feedback to validate the prototypes and completed an at-home trial with the prototypes to study the effects of vibration therapy on PLP when administered via a vibration liner, addressing the third project aim. Participant feedback and an assessment of contact between the liner and socket were used to evaluate the check socket’s fit. To assess whether the suction of the check socket was maintained, a test was performed wherein the participant donned the socket over the prototype liner and attempted to pull their residual limb out of the socket while a researcher held the socket steady. During the four-week at-home trial, participants were instructed to use the vibration liner in the event of PLP. They were asked to only use the liner when sitting or lying down to comply with the approved study protocol. They were also asked to record the date, start time, and end time of using the liner and rate their pain intensity before and after using the liner on the 11-point Numeric Pain Rating Scale, where 0 indicates no pain and 10 indicates the worst pain imaginable [18]. A one-tailed paired sample *t*-test (*α* = 0.05) was used to compare pain intensity before and after using the vibration liner and determine whether to reject the null hypothesis. Qualitative feedback was collected during a post-at-home-trial semi-structured interview consisting of 10 pre-written questions aimed at obtaining feedback on three primary themes: the usability and design of the vibration liner, the effect of vibration therapy on PLP, and the effect of vibration therapy on PLS. Responses were transcribed during each interview. Following the interviews, the transcriptions were reviewed for commonalities related to the primary themes.

## 3. Results

### 3.1. Bench Testing

The prototype used for the donning and doffing test had two embedded motors. After 70 cycles of donning and doffing, the Sil-Poxy covering one motor began to show signs of wearing through, but the motor itself was unaffected. After 400 cycles, the Sil-Poxy covering this motor had worn off, and the edge of the motor began to poke out of the liner. The Sil-Poxy covering the second motor remained intact and showed no signs of wear or tears. The conductive hook-and-loop showed no signs of wear and remained securely embedded in the liner. Both motors were tested for functionality and passed each time. Figure 4 shows the motors before and after this test.

The liners were cleaned with a pH-neutral prosthetic cleanser according to the instructions from the cleanser manufacturer, and no damage was observed. After 35 cycles, the SC and SAC trials resulted in no tearing and minimal peeling of the copper tape strips, and the SAB trials resulted in a similar degree of peeling and slight tearing of the copper tape strips. The copper tape strips from all trials were functional when tested after the cleaning protocols. Figure 5 shows the results of this test.

### 3.2. Clinical Testing

Two individuals with transfemoral limb loss participated in the data collection. The elements added to the liners and check sockets to create the prototypes did not affect the fit or suspension of the sockets. The participants said the sockets fit comfortably, and the vibration motors were not poking them through the liner, nor did they notice the motors being pressed against their skin. After donning the vibration liner and check socket, the gray band around the liner, used for vacuum seal, had no wrinkles or ripples, indicating that it was making good contact with the socket (Figure 6). Additionally, each participant’s residual limb evenly contacted the socket to the same degree as before the modifications. The suspension continued working as intended, with each participant unable to pull their residual limb out of the socket when tested.

During the four-week at-home trial, the participants reported a reduction in pain intensity after using the vibration liner. One participant experienced PLP once throughout the four weeks, and another participant experienced PLP an average of three times per week. Before using the vibration liner, the participant’s reported pain intensity ranged between 3 and 6 (mean 4.33) with an outlier at 2. The median pain level was 4.5 (IQR 4–5). After using the vibration liner, the participant’s reported pain intensity ranged between 0 and 1 (mean 0.75). The median pain level was 1 (IQR 0.25–1). Statistical comparison of the pre- and post-vibration pain level on the latter participant’s data indicated a significant (*p* < 0.001) improvement of the average pain intensity from 4.33 to 0.750 on an 11-point scale (Figure 7).

The participants provided qualitative feedback on vibration therapy and the vibration liner prototype. Their feedback validated that the embedded vibration motors and electrical contacts did not poke their skin or interfere with rolling the liner to don and doff it:

*“The liner was comfortable to wear. It didn’t take any more or less effort to put on. You haven’t disrupted the donning or doffing of the liner.”* (Participant 1)

*“I try to be a little more careful with the vibration liner, but it isn’t more difficult to put on. Not at all. There are no more steps to putting it on compared to my other liner.”* (Participant 2)

Having vibration applied to the residual limb was described as relaxing and soothing. The vibration provided stimulation to the muscles and nerves of the residual limb. One participant also described a relaxation of his PLS:

*“The vibration is a dual sensation. I feel the vibration on my residual limb where it’s vibrating, but I also definitely feel the vibration in my foot. The pain is in my foot, but my residual limb is still being contacted by the liner and massaged by the vibration.”* (P1)

*“The vibration feels like it’s extending my foot. My foot can feel tense, but with the vibration, my toes are not scrunched up. They feel totally relaxed. It’s like being at the spa and getting a massage. It feels phenomenal.”* (P2)

Both participants expressed experiencing pain relief from the vibration liner. The feeling of PLP when using the vibration liner was described as dulled and numbed:

*“The vibration soothes the pain. It doesn’t take it away, but it definitely lightens the pain. It dulls it.”* (P1)

*“The vibration mellows out the pain. It numbs the nerves. The pain may still be there, but it’s very mild and feels tingly. A good kind of tingly that’s not painful.”* (P2)

Both participants expressed they would like a stronger vibration and suggested a stronger vibration may improve the reduction in pain intensity. They both mentioned having difficulty aligning the contacts and maintaining contact when doffing:

*“When I make the wrap tighter, it makes the vibration stronger, and that feels better. It feels good with any vibration, but it feels great when it is stronger.”* (P1)

*“Stronger vibration would have felt better and maybe knocked the pain down some more. Making good contact [with the motors] was the toughest part, especially while you’re in pain in the middle of the night in the dark.”* (P2)

## 4. Discussion

The results of the bench testing and clinical testing demonstrate that the project aims and design criteria were successfully met, as summarized in Table 3. The method of embedding motors and wires was a feasible method to withstand the average six-month lifespan of prosthetic liners and cleaning the liner with a prosthetic-safe cleanser. While the Sil-Poxy covering one motor wore through and exposed the motor during the bench test, this was likely due to that layer of Sil-Poxy being initially thin. For the motor that was covered with a thicker layer of Sil-Poxy during the bench test, no wear to the Sil-Poxy was observed, and the motor remained securely embedded in the liner. Applying a thicker layer of Sil-Poxy when embedding the motors in the liner could address the wear seen with the one motor. Motors in subsequent prototypes were covered with a thicker layer of Sil-Poxy, and no wear or tearing was observed. These subsequent prototypes were used by the participants in the lab while preparing the prototypes for the at-home trial and were sent home with the participants during the at-home trial. The thicker covering of Sil-Poxy on the exterior of the liner did not affect the comfort or ease of use of the liner, as both participants said the prototype liners were comfortable and no more difficult to don and doff compared to their personal liners. Cleaning the socket and liner was unaffected by the addition of the prototype elements. An abrasive cleaning method, such as using a bristle brush, would result in some peeling and tearing of the copper tape, but the participants’ current socket cleaning methods would only cause minimal peeling at the edges of the copper tape; the adhesion and functionality of the copper tape remained unaffected. For future prototypes, a waterproof adhesive can be added over the copper tape on the inside of the socket to reduce the wear to it from cleaning. Participant feedback confirmed that the liner and socket were comfortable, and the prototype components did not interfere with the suspension inside the socket.

The results from the at-home trial add to previously collected data that support using vibration therapy for PLP. After using the vibration liner, the participant’s pain intensity was rated an average of 3.5 points lower than the initial pain intensity. A study assessing the clinical importance of changes in chronic pain intensity when measured on an 11-point numeric pain rating scale concluded that a 2-point reduction from the initial pain rating represented a clinically important improvement in pain level [19]. The results of the at-home trial represent a clinically important and statistically significant improvement in pain level and support the alternative hypothesis that vibration therapy administered via a vibration liner reduces the intensity of PLP.

Given the large prevalence of PLP and the somewhat intractable nature of the problem, a solution, such as the vibration-motorized liner, that is non-invasive, non-pharmacological, and compatible with conventional prosthesis technology and use can have a substantial impact on the quality of life of people with limb loss. Beyond this, some of the technology that has been developed and tested in this project may hold potential for unrelated applications that require the instrumentation of prosthetic liners. Measuring data, such as temperature, moisture, friction, or contact pressure at the limb–prosthesis interface is quite helpful in identifying fitting problems and possible optimization interventions. However, the difficulty (and cost) of not only implementing the respective sensors but also connecting them to power supply and data loggers has rendered most of the associated attempts infeasible for any wider application beyond research projects. It should be possible to address some of the underlying technical problems with the embedding and wiring methods described here. It makes no substantial difference whether it is a vibration motor, as in our project, a temperature sensor, or any other small sensor that is connected without the need for wiring through the flexible liner material. Even the implementation of actuators (e.g., for volume control) in a future “robotized” liner generation should be able to utilize this approach.

A limitation of this study is the small number of participants. Since pain is subjective and the occurrence of PLP is unpredictable, this study was limited in the amount of data collected. A second limitation was the inability to blind the participants to the intervention. The participants’ enthusiasm for the intervention and participation with the previous student teams may have been a source of potential bias. The prototypes were tested with transfemoral amputees, so the results cannot be generalized to amputees with other levels of amputation.

Further development to make the vibration stronger with and without the socket should be undertaken. When testing the vibration liner within the socket, the vibration became muted, so methods to reduce how much the socket mutes the vibration should be explored. Additional development regarding aligning the contacts of the liner and the conductive wrap is also necessary. The current design requires the copper tape on the conductive wrap to make total contact with the conductive hook-and-loop of the liner and does not allow for much deviation and misalignment by the participant during set up. Additionally, due to the conical shape of the residual limb of a transfemoral amputee, the conductive wrap tended to slide toward the distal end of the participant’s residual limb during use, thus losing contact with the conductive hook-and-loop. To address this feedback, methods to improve initial alignment and maintain alignment during use should be investigated. A potential method could include securing the conductive wrap to the conductive hook-and-loop to indicate to the user that the connection has been made successfully and prevent the conductive wrap from sliding during use. Since the statistical power of this study is limited by a small number of participants, future work should increase the sample size. The results from this study suggest that administering vibration therapy via a vibration liner significantly reduces PLP, so increasing the statistical power of the results will further support vibration therapy as an effective therapy option and improve the generalizability of the findings. Future work should include transtibial amputees, and adjustments to the socket prototypes may be necessary to integrate the prototypes with transtibial socket suspension methods. Additionally, longer at-home trials should be conducted to capture data from more PLP occurrences in a diverse population and to study any potential long-term effects of vibration therapy on PLP.

## 5. Conclusions

Results from this pilot study support the use of vibration therapy to reduce the intensity of PLP. The current prototype demonstrates the potential to achieve prosthesis-compatible vibration therapy by improving user-friendliness and durability compared to earlier prototypes, in addition to being compatible with a prosthesis socket without interfering with suspension.

## 6. Patents

B. Perry, G. Fiedler, and K. C. Quinn, ‘Vibratory Devices for Phantom Limb Pain’, Patent US11577045B2, 14 February 2023 [13].

K. Roehrich, G. Fiedler, and N. Stivala, ‘Conductive Connection Systems for Use in Connection with Prosthesis Liners’, Provisional Patent Application 63/557,042, 2024 [20].

## Figures and Tables

**Figure 1 sensors-24-05026-f001:**
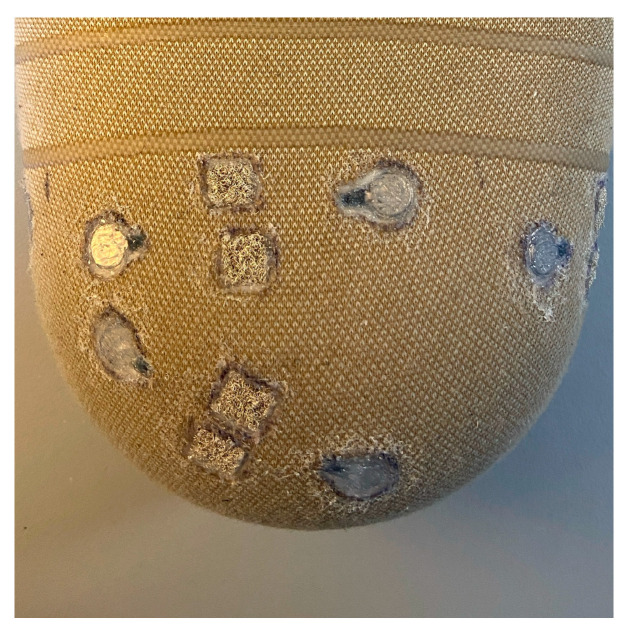
Posterior view of the final vibration liner prototype. One pair of motors is located three centimeters from the distal end of the femur, while four pairs of motors are located eight to ten centimeters above the distal end of the femur and spaced circumferentially around the liner. Each motor is embedded in the silicone and sealed with Sil-Poxy. The wires from each motor terminate under the conductive hook-and-loop pads that serve as contact points when providing power to the liner.

**Figure 2 sensors-24-05026-f002:**
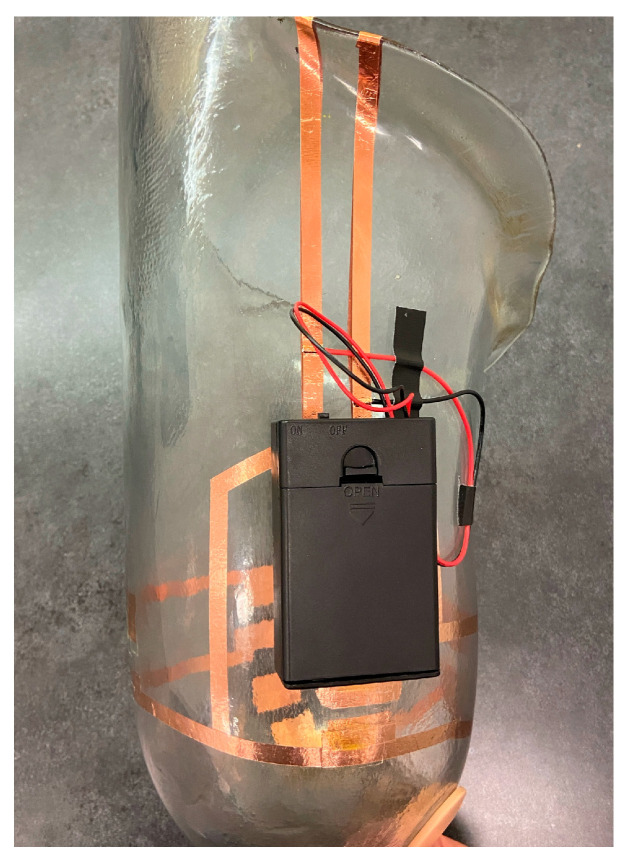
Anterior view of the final socket prototype with a battery holder attached. The arrangement of the copper tape aligns with the contact points on the liner when the socket is donned.

**Figure 3 sensors-24-05026-f003:**
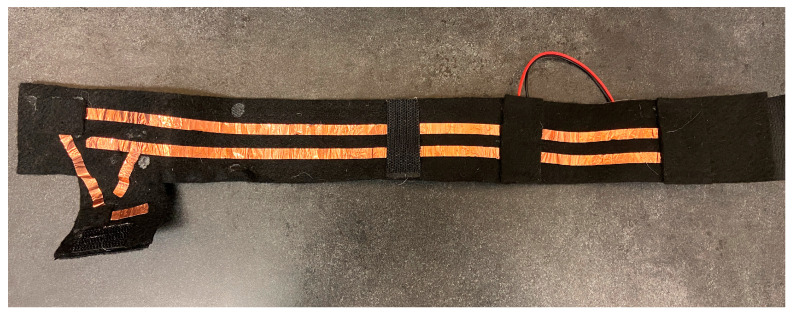
Conductive wrap to provide power to the vibration liner without donning the socket. The arrangement of the copper tape aligns with the contact points on the liner when wrapped around the liner.

**Figure 4 sensors-24-05026-f004:**
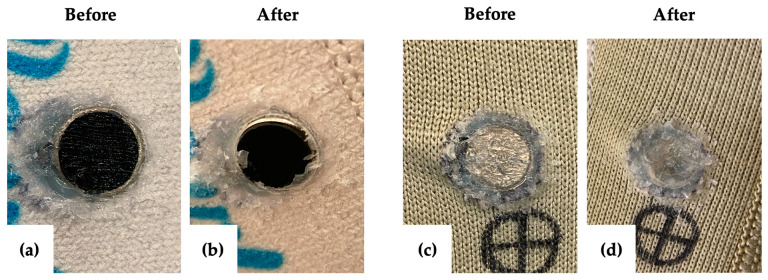
Results of the donning and doffing test. Images (**a**,**c**) were taken before the test, and images (**b**,**d**) were taken after the test. The Sil-Poxy layer in image (**b**) has worn off but remains intact in image (**d**).

**Figure 5 sensors-24-05026-f005:**
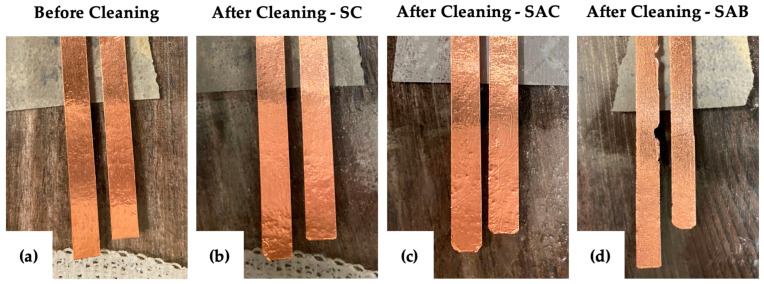
Results of the socket cleaning test. (**a**) Copper tape before cleaning, followed by images of copper tape after cleaning with (**b**) soap and cloth (SC); (**c**) soap, cloth, and isopropyl alcohol (SAC); and (**d**) soap, isopropyl alcohol, and a scrub brush (SAB). All trials resulted in minimal peeling, and only the trial using the brush resulted in slight tearing of the copper tape.

**Figure 6 sensors-24-05026-f006:**
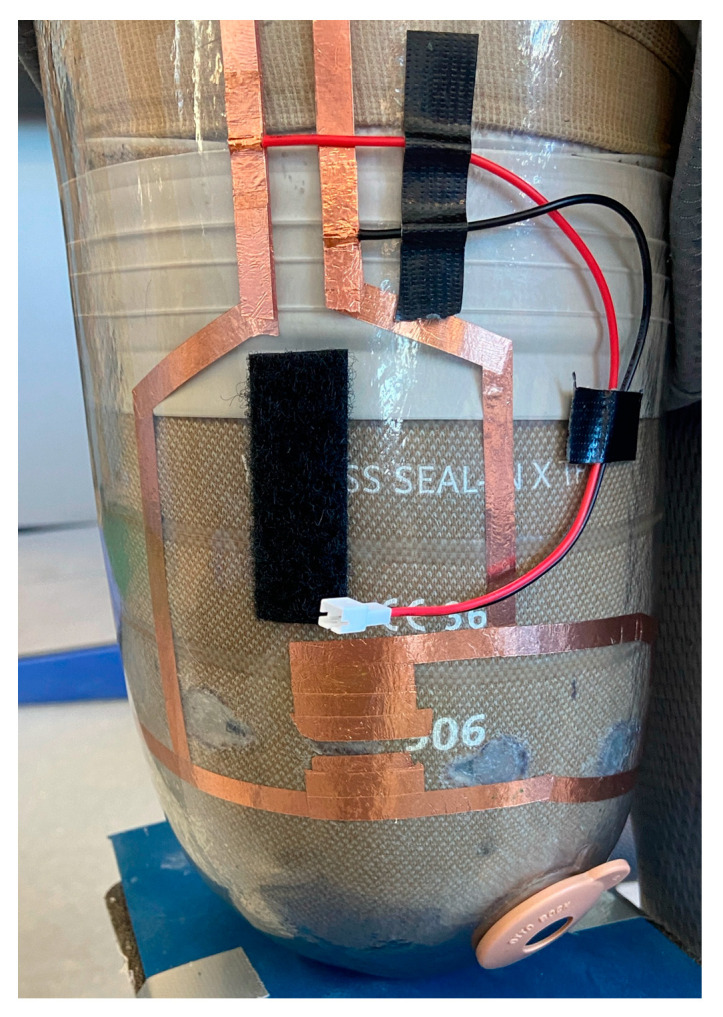
Check socket prototype donned over vibration liner prototype. The liner evenly contacts the socket, and suction is maintained. The copper tape is in alignment with the conductive hook-and-loop of the liner.

**Figure 7 sensors-24-05026-f007:**
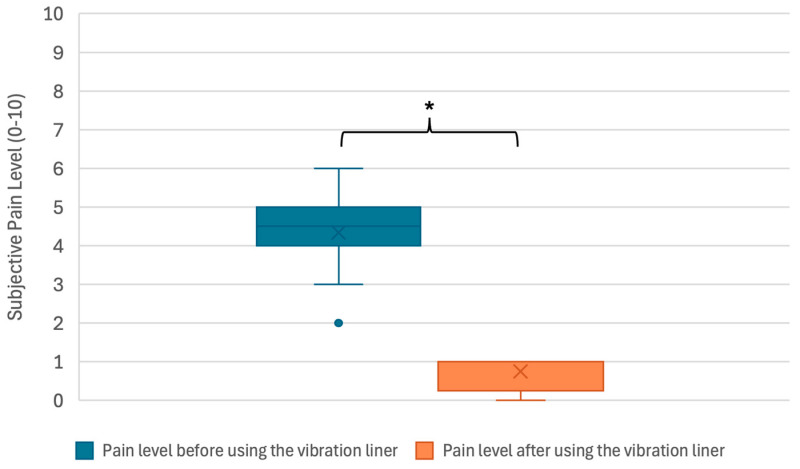
Box plot showing the distribution of participants’ reported pain intensity (blue) before using the vibration liner and (orange) after using the vibration liner. Before using the vibration liner, the pain intensity ranged between 3 and 6 (mean 4.33; median 4.5; IQR 4–5). After using the vibration liner, the pain intensity ranged between 0 and 1 (mean 0.75; median 1; IQR 0.25–1). ***** Shows a significant decrease in pain level after using the vibration liner (*p* < 0.001, *n* = 12).

**Table 1 sensors-24-05026-t001:** Demographic information of study participants.

Variables	Study Population (N = 2)
Age	
Mean	56 years
Gender	
Male	2
Female	0
Height	
Mean	175.26 cm
Body Mass Index	
Mean	21.5
K-Level	Level 3
Prosthesis Experience	10+ years each

**Table 2 sensors-24-05026-t002:** Description of the design criteria for the prototypes based on the project aims.

Aim	Criteria	Rationale	Previous Attempt(s)
Aim 1: Develop a method of powering an instrumented prosthesis liner via a power source located on the exterior of the socket without affecting the comfort and fit of the socket	The prototype must not interfere with socket suspension.	Maintaining suction is crucial to ensure the prosthesis is secure and stable during ambulation.	Prototypes were too thick or too delicate to fit inside the socket.
Aim 2: Develop a method of securely embedding vibration motors into a liner so they remain embedded during regular use of the liner	Any added electronic components must withstand repeated donning and doffing of the liner.	Liners provide protection, cushion, and suspension within the socket and need to be flexible and elastic roll during donning and doffing.	Wires embedded within silicone liners became disconnected where spliced (2022 prototypes)
	Any components embedded within a liner need to be slim and must not poke the skin.	Liners typically taper from a thickness of 7 mm at the distal end to 2.5 mm at the proximal end [16].	
Aim 3: Study the effects of vibration therapy on the intensity of phantom limb pain when delivered via a vibration liner, investigating the hypothesis that vibration therapy delivered via a vibration liner would reduce the intensity of PLP	In addition to functioning inside a socket, the prototype must also function without a socket.	Participants most-often experience PLP early in the morning or in the middle of the night when they are not wearing their socket.	Difficult to set up when experiencing PLP (2022 prototype) Battery life was too short (2023 prototype)
	The prototype must have rechargeable or easily replaceable batteries.	Participants often experience PLP lasting multiple hours.	500 mAh Lithium-Polymer battery with a 45 min run time (2023 prototype)

**Table 3 sensors-24-05026-t003:** Summary of criteria following bench testing and clinical testing.

Criteria Met?	Criteria	How Criteria Were Met
Yes	The prototype must not interfere with socket suspension.	Suction was maintained, and the socket remained securely suspended on the participant’s residual limb.
Yes	Any added electronic components must withstand repeated donning and doffing of the liner.	The prototype passed a bench test of 400 donning and doffing cycles.
Yes	Any components embedded within a liner need to be slim and must not poke the skin.	Prototype elements did not poke through liner, nor did they cause discomfort to the wearer.
Yes	In addition to functioning inside a socket, the prototype must also function without a socket.	Vibration liner was functional after donning the socket and using the battery pack on the socket. Vibration liner was functional after donning the conductive wrap and using the battery pack with the conductive wrap.
Yes	The prototype must have rechargeable or easily replaceable batteries.	The prototype used a battery pack that contained three AA batteries.

## Data Availability

The original contributions presented in the study are included in the article, further inquiries can be directed to the corresponding author/s.

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
