# Peer review of "Robotic-Enhanced Prosthetic Liners for Vibration Therapy: Reducing Phantom Limb Pain in Transfemoral Amputees"

_sensors, 2024, doi:10.3390/s24155026_

Round 1
Reviewer 1 Report
Comments and Suggestions for Authors
Clarification of Figures: The figures that include multiple outcomes (e.g., Figure 4 and Figure 5) should be clearly labeled to enhance readability and facilitate easy observation. This will help readers to distinguish between different results presented within the same figure.
Participant Demographics: There is an absence of detailed participant information, which is crucial for the study's context. The inclusion of data such as the number of male and female participants, their age range, average height, and body mass index would provide a more comprehensive understanding of the study's population.
Enhanced Descriptions: The description of the "Sil-Poxy covering one motor wearing off and exposing the motor" requires further clarification. It would be beneficial to know the frequency of this occurrence and the underlying causes. Additionally, if the thinness of the material contributed to its wear, it's pertinent to discuss whether a thicker covering could affect comfort and how a balance between sensory comfort and material durability might be achieved.
Consistency in Rating Scales: The standard numerical rating scale for subjective evaluations is typically set to 10 points. However, this research opts for an 11-point scale to assess pain levels (Line 273). This deviation from the norm warrants an explanation. Furthermore, Figure 7 uses a 10-point scale to measure pain intensity, which introduces inconsistency. Clarification on the rationale behind these choices and the implications for the study's findings is necessary.
Comments on the Quality of English LanguageThe clarity of certain descriptions should be enhanced to improve the overall readability of the article
Reviewer 2 Report
Comments and Suggestions for Authors
This manuscript proposes a prosthesis liners with integrated vibration motors to administer vibration therapy for phantom limb pain. The related results are provided to illustrate the effectiveness of the proposed vibration therapy scheme. Hence, the manuscript is interesting and the presentation is well-written.
The author can refer to the following comments, and improve the contribution and presentation in the revision.
(1) What is the different of the proposed vibration therapy from the other method, especially in reducing phantom limb pain?
(2) How to evaluate the performance of the designed vibration therapy framework? Meanwhile, how to evaluate the vibration therapy performance in this study?
(3) Since some patient have different motion styles and features, the designed scheme should consider more information about individual feature.
(4) How about the vibration liner performance affect the therapy effect in different motion posture?
(5) Some new references also discussed wearable control of lower limb similar to the author’s work, such as output constrained control of lower limb exoskeleton based on knee motion probabilistic model, and human-exoskeleton coupling dynamics in the swing of lower limb. The authors may give more descriptions about related work.
(6) There exist many typos and grammar errors in text. Please carefully check all the presentations in the revision.
Comments on the Quality of English LanguageThere exist many typos and grammar errors in text. Please carefully check all the presentations in the revision.
Reviewer 3 Report
Comments and Suggestions for Authors
The manuscript discussed prosthesis liners with integrated vibration motors to administer vibration therapy for phantom limb pain. The topic is of interest for this journal and some comments are provided for improvements.
1. The technical specifications of the vibration therapy used in the liners are not clearly detailed. For instance, what were the frequency and amplitude of the vibrations, and how were these parameters determined? Please provide clarification.
2. Participants mentioned that aligning the contacts could be easier. Could you provide more details on the current design of the contacts and any potential modifications you are considering to address this feedback? Please clarify.
3. The conclusion mentions that abrasive cleaning methods could cause some peeling of the copper tape. This is a practical issue, and how to address it? Please clarify.
Comments on the Quality of English LanguageMinor changes.
Reviewer 4 Report
Comments and Suggestions for Authors
The study presents the design and creation of a prosthetic liner embedded with vibration motors intended for vibration therapy to alleviate PLP.
This study provides clear descriptions of the materials used and the experimental procedures. This level of detail ensures that the study can be replicated by other researchers.
The research design is detailed, with a clear structure that includes both bench testing and clinical testing. The inclusion of real participants makes the research more meaningful.
-- The sample size in the study is relatively small, which may limit the generalizability of the findings. To strengthen the validity of the results, it is recommended to increase the number of participants. A larger sample size will enhance the statistical power of the study and provide a more robust evaluation of the effectiveness of vibration therapy on PLP.
The manuscript is well-written and easy to follow. However, some sections could benefit from additional detail or clarification.
-- Specify the precise values denoted by each identifier in Figure 7.
-- The academic language used in Table 2 is not standardized.
The study demonstrates a commendable level of humanitarian concern. The authors have shown a deep understanding of the physical and psychological challenges faced by PLP sufferers and have made efforts to improve their quality of life through innovative therapy methods.
